# Colonization of the Gastrointestinal Tract of Chicks with Different Bacterial Microbiota Profiles

**DOI:** 10.3390/ani13162633

**Published:** 2023-08-15

**Authors:** Laura Franco, Martine Boulianne, Eric Parent, Neda Barjesteh, Marcio C. Costa

**Affiliations:** 1Department of Veterinary Biomedical Sciences, Faculté de médecine vétérinaire, Université de Montréal, Saint-Hyacinthe, QC J2S 2M2, Canada; laura.franco@umontreal.ca; 2Department of Clinical Sciences, Faculté de médecine vétérinaire, Université de Montréal, Saint-Hyacinthe, QC J2S 2M2, Canada; martine.boulianne@umontreal.ca (M.B.); eric.parent.1@umontreal.ca (E.P.); 3Department of Pathology and Microbiology, Faculté de médecine vétérinaire, Université de Montréal, Saint-Hyacinthe, QC J2S 2M2, Canada; neda.barjesteh@zoetis.com; 4Global Companion Animal Therapeutics, Zoetis, Kalamazoo, MI 49007, USA

**Keywords:** microbiota transplant, FMT, microbiome, chicken, poultry

## Abstract

**Simple Summary:**

This study investigated the effects of transplanting different bacterial profiles into newly hatched broiler chicks. We hypothesized that chicks receiving gut bacteria from organic hens would have a distinct microbiota compared to those receiving bacteria from conventionally raised broilers. It was found that the chicks developed a gut microbiota similar to that of the donors from whom they received their bacteria. Over time, the microbiota from a control group resembled the microbiota of conventionally raised broilers, but the bacteria received from organic hens remained different from the other groups until 42 days of live. The chicks that received bacteria from conventionally raised broilers showed higher inflammation due to Eimeria infestation. The study concluded that gut bacteria transplantation can persistently colonize chicks. This information could be used in the future to select species of importance that are more adapted to the intestinal tract of chickens, and it highlights the importance of the strict screening of donors used in microbiota transplantation.

**Abstract:**

This study aimed to investigate the consequences of early-life microbiota transplantation using different caecal content sources in broiler chicks. We hypothesized that chicks receiving at-hatch microbiota from organic hens would harbour a distinct microbiota from chicks receiving industry-raised broiler microbiota after six weeks of age. Three hundred Cobb broilers eggs were randomly assigned to one of four groups according to the caecal content received: organic laying hens (Organic); autoclaved caecal content of organic laying hens (Autoclaved); conventionally grown broilers (Conventional); and sterile saline (Control). caecal microbiota transplantation was given by gavage on day 1. Ten birds/group were euthanized on days 2, 7, 14, 28, and 42. The caecal tonsils and contents were collected for cytokines and microbiota analyses. The microbiota from chicks receiving live inocula resembled the donors’ microbiota from day seven until day 42. The microbiota composition from the chickens who received the Organic inoculum remained markedly different. Starting on day 7, the Organic group had higher richness. Simpson and Shannon’s indices were higher in the Conventional group on days 2 and 7. Chickens in the Conventional group presented higher production of IL-1β and IL-6 in plasma on days 2 and 28, increased IL-6 expression in the caecal tonsils at days 7 and 42, and increased IL-12 expression on day 7. However, the Conventional group was infected with *Eimeria* spp., which likely caused inflammation. In conclusion, microbiota transplantation using different microbiota profiles persistently colonized newly hatched broiler chicks. Future studies evaluating the importance of microbiota composition during infections with common enteropathogens are necessary. This study also highlights the need for a strict screening protocol for pathogens in the donors’ intestinal content.

## 1. Introduction

The intestinal microbiota composition can impact the development and maturation of the gastrointestinal tract (GIT) and immune system [1]. The essential role of the gut microbiota in maintaining animal and human health is well established [2,3,4,5], as intestinal bacteria are responsible for the breakdown of foodstuffs, proper nutrient absorption, and growth [6]. Indeed, the intestinal microbiota exerts an essential role in fiber digestion, further improving feed utilization, volatile fatty acid production and, consequently, weight gain and growth performance indicators [7]. For instance, *Bacteroides fragilis*, *Ruminococcus* sp., and *Lactobacillus coleohominis* are butyrate producers and have been associated with better daily weight gain in chickens [7].

Each animal species has coevolved in nature with its microbial symbionts, providing both host and microbes survival advantages. Host evolutionary history is thought to be a driving factor in determining the composition of intestinal bacteria [8,9,10,11,12]. The consequences of colonization with altered microbiota profiles due to human intervention are just beginning to be investigated. In nature, wild animals have richer and more diverse microbiota compared to domestic animals, and the same is true for birds [13]. Although some studies suggested that a rich and diverse caecal bacterial community can positively affect performance [14,15], other studies have found that certain practices to increase weight gain (i.e., low doses of antibiotics) are, in fact, associated with decreased diversity in the intestinal microbiota [16]. In modern poultry production practices, fertilized eggs and the resulting progeny do not have maternal care nor contact [17]. Strict hygienic measures, such as environmental disinfection, facility-restricted access, sanitation of eggshells at the hatchery, and lack of contact with adult animals, are likely to influence intestinal colonization by pioneer bacteria [18,19]. Still, more research is needed to investigate the importance of eggshell microbiota on the colonization of chicks [20].

Over the last few decades, intensive selection has resulted in chickens having the lowest feed conversion ratio amongst common agricultural animals, accompanied by early maturation of the digestive system. Microbial succession and early GIT colonization are essential drivers of host health [21,22,23]. Colonization of the chicken’s intestinal microbiota begins immediately post-hatch, if not before. It is influenced by intrinsic (e.g., host genetics and transfer at the time of the egg formation) and external (e.g., diet and environmental microbes) factors [24], achieving specific stability after the first three weeks of life [25]. In the presence of pathogenic bacteria at hatch, the chick’s GIT represents an empty ecological niche which allows rapid pathogens colonization, and Proteobacteria, a phylum commonly associated with intestinal diseases and inflammation, has been reported to be a major component of the neonatal chick’s microbiota [21,26,27,28]. However, this initial colonization’s early- and late-life consequences remain to be investigated [29,30]. Noteworthy, the life cycle of chickens is relatively short, which makes this maturation window of the GIT microbiota even more relevant.

Metabolic disorders, high immune challenges, and increased susceptibility to pathogens accompany intensive production systems. Newly hatched chicks primarily depend on innate immune responses until their gut is colonized with bacteria. Consequently, early exposure to active substances and beneficial bacteria (pre- and post-hatch) during the first two weeks of life might be crucial for developing the immune system [31]. Experiments with germ-free birds have shown that the intestinal microbiota influences intestinal T-cell repertoire and cytokine expression [32,33]. Furthermore, certain commensal bacteria can increase T_h_17 cells in the lamina propria [34] and the secretion of IL-17A, a pro-inflammatory cytokine that plays a signalling role in several diseases [35]. IL-17 is an essential host defense mediator against important pathogens, such as Marek’s disease virus, *Cryptosporidium baileyi*, and *Eimeria* spp. [36,37,38].

In human medicine, faecal microbiota transplants (FMT) have been used to successfully modulate and restore the intestinal microbiota, such as in cases of recurrent *Clostridioides difficile* infections [39,40]. Similarly, in chickens, administration of the faecal microbiota from healthy adults has been used to colonize newly hatched chicks, increasing resistance against *Salmonella* spp. [41]. Inoculating the eggshells with caecal contents from highly and poorly feed-efficient donor chickens has been shown to reduce bird-to-bird variation in microbiota composition, but it did not impact feed efficiency in the growing chicks [26]. Therefore, microbiota manipulation might be a powerful and helpful tool with which to modulate the immune system and improve performance, but further studies are needed to better understand and refine those methods.

This study aimed to investigate the consequences of early-life microbiota transplantation in broiler chicks colonized with different sources of caecal contents. The hypothesis was that the highly diverse microbiota of organic laying hens could colonize chicks, and the chicks would harbour a different microbiota and immunity compared to those colonized with industry-raised broiler chicken’s microbiota and to negative controls later in life. Furthermore, the study tested the hypothesis that a caecal solution containing only metabolites and autoclaved unviable bacteria would impact the microbiota and immunity of broiler chicks.

## 2. Materials and Methods

### 2.1. Ethical Statement

All experimental procedures were approved by the Comité d’éthique de l’utilisation des animaux (CÉUA) of the Université de Montréal (Animal Utilization Protocol #18-Rech-1953). The study was conducted following the guidelines of the Canadian Council on Animal Care (CCAC).

### 2.2. Experimental Design

Three hundred broiler embryonated eggs were acquired two days before hatching from Cobb-Vantress Inc. Eggs were randomly assigned to one of four groups (*n* = 75) according to the origin of the caecal content that the chicks received: free-range organic laying hens (Organic); autoclaved caecal content of organically raised laying hens (Autoclaved); conventionally grown broiler chickens (Conventional); sterile saline solution (Control).

### 2.3. Donor Selection and Inocula Preparation

Twenty free-range, 86-week-old, brown Lohmann laying hens were obtained from an organic farm (Saint-Ours, QC, Canada). They were fed a standard diet that did not contain antibiotics and had open access to pasture and foraging behaviour. Birds were transported to the Centre de recherche avicole (Saint-Hyacinthe, QC, Canada) and were electrically stunned and euthanized by a section of the jugular vein within 20 s of stunning. The contents of each caecum were immediately obtained by milking the intestinal compartment. Samples from all laying hens were homogenized and pooled together in a jar. The total content was divided into two inocula using ~20 g of caecal contents from organically reared hens for each, which were diluted in 500 mL of sterile saline solution (0.9 *w*/*v*) each. One of these inocula was autoclaved with pressurized saturated steam at 121 °C for 15 min to sterilize the solution.

Parallelly, the caeca of ten 6-week-old conventionally raised broiler chickens were collected from a commercial slaughterhouse on the same day (Saint-Damase, QC, Canada). The caeca were refrigerated and transported to the Faculté de médecine vétérinaire (Saint-Hyacinthe, QC, Canada), where ~20 g of caecal contents of ten birds were diluted in 500 mL of sterile saline solution (0.9 *w*/*v*). All the inocula were sealed and stored at 4 °C for 12 h until the inoculation the following morning.

### 2.4. Caecal Microbiota Transplantation

Upon arrival, embryonated eggs from all four treatment groups were transferred into two clean incubators (NatureForm^®^) set at 37 °C and with an average relative humidity of 73%. The transplantation was performed by brushing 50 mL of the respective inoculum or sterile saline solution onto the eggshells and spraying 100 mL on the wood shaving of the corresponding pen. Within the first 12 h after hatching, chicks received 200 µL of their respective inoculum or sterile saline solution via gavage using a 1 mL sterile syringe and transferred to their separate pens. Finally, 150 mL of the inocula were added to 850 mL of drinking water fountains for the first-day post-hatch. Samples were collected from each inoculum, as well as from the drinking fountains, for microbiota analysis.

Birds of each group were further split into two temperature-controlled rooms; each room had four ground-rearing pens, one for each treatment group. Therefore, each treatment group had two replicates. Birds were fed a starter (days 0–13), a grower (days 14–27), and a finisher diet (days 28–42) (La Coop Natur-aile Feed) ad libitum and had constant access to water.

### 2.5. Sample Collection

On days 2, 7, 14, and 28 post-hatching, ten birds from each treatment group (five from each room) were euthanized via exsanguination through the jugular vein after losing consciousness from CO_2_ asphyxia. The remaining birds in each group were euthanized on day 42 post-hatching (*n* = 42). The birds’ body weight was recorded post-asphyxiation. The caecal tonsils and two grams of fresh caecal contents per bird were collected and stored in 2 mL tubes at −80 °C. After collection, caecal tonsils samples were immediately stored in RNAlater™ (QIAGEN, Toronto, ON, Canada). Blood was collected (2.0 mL/bird) from the jugular vein and stored on ice (one hour) until centrifugation (4 °C, 2500 rpm, 20 min) for plasma separation. Plasma was aliquoted into 1.5 mL tubes and stored at −20 °C until further analysis.

### 2.6. DNA Extraction and 16S rRNA Sequencing

Total DNA was extracted from the caecal content samples using the glass bead-based extraction kit DNeasy PowerSoil^®^ Kit (QIAGEN, Toronto, ON, Canada) following the manufacturer’s protocol. DNA quantification and quality were assessed via spectrophotometry using a NanoDrop™ 1000 (Thermo Fisher Scientific, Waltham, MA, USA), and DNA was stored at −20 °C until analysis. The V4 hypervariable region of the 16S rRNA gene was PCR-amplified using the 515F (5′-GTGCCAGCMGCCGCGGTAA-3′) and 806F (5′-GGACTACHVGGGTWTCTAAT-3′) [42]. Paired-end sequencing of amplicons was performed with an Illumina MiSeq (Illumina, San Diego, CA, USA) platform for 250 cycles from each end at the McGill Génome Québec Innovation Centre (Montréal, QC, Canada) [43].

### 2.7. Sequence Analysis

DNA sequences were analyzed using the software Mothur (version 1.47.0) [44]. Data analysis was performed as described previously [45]. Briefly, reads containing more than 300 bp and more than eight homopolymers and any ambiguity were excluded. Good-quality reads were aligned to the SILVA bacterial database. Chimeras were removed with the v.search algorithm, and taxonomy was assigned to each sequence using the Ribosomal Database Project bacterial taxonomy classifier. Reads with 96% similarity were clustered, reads appearing only once or twice were excluded, and reads belonging to the same genus were clustered (phylotypes).

Alpha-diversity was characterized by richness (the total number of observed taxa and the Chao index) and diversity (the Simpson’s and Shannon indices). Beta-diversity was investigated by employing community composition using the Jaccard index (accounting only for the presence or absence of each taxon) and the Yue and Clayton index (accounting for the presence and relative abundance of each taxon in the community). Subsampling using the smallest number of reads obtained in a sample was used to standardize non-uniform samples to avoid introducing bias into the analysis.

### 2.8. RNA Extraction, cDNA Synthesis and Quantitative PCR

Total RNA was extracted from the caecal tonsils samples using TRIzol Reagent (Invitrogen™, Carlsbad, CA, USA). cDNA was generated using SuperSript™ VILO™ cDNA Synthesis Kit (Invitrogen™, Carlsbad, CA, USA) following the manufacturer’s instructions. The gene expression of Interleukin-4, Interleukin-6, Interleukin-10, and Interleukin-12 was evaluated using a ViiA™ 7 Real-Time PCR System (Applied Biosystems™, Foster, CA, USA) with PowerUp™ SYBR™ Green Master Mix (Applied Biosystems™, Foster, CA, USA). Data were expressed as an n-fold difference relative to the expression of the internal control housekeeping gene, β-actin. The primer sequences are shown in Appendix A.

### 2.9. Enzyme-Linked Immunosorbent Assay

Chicken specific-antigen ELISA kit assessed levels of Interleukin-1β, Interleukin-6, and Interleukin-8 in blood serum on days 2, 7, and 28 using pair-matched antibodies from ABClonal Inc. (Woburn, MA, USA), according to the manufacturer’s recommendations in triplicates. Once the reaction was completed, the optical density was measured at 450 nm on a Fusion Microplate reader (Packard BioScience, Meriden, CT, USA).

### 2.10. Statistical Analyses

Statistical comparisons were performed using GraphPad Prism software (version 9.4). One-way ANOVA was used with Tukey’s multiple comparison tests to compare cytokine production (ELISA), gene expression (RT-qPCR) and average body weight gain between the four treatment groups at different sampling days. A mixed-effect model with Tukey’s multiple comparison tests was used to compare the alpha-diversity indices of all the treatment groups. Unless mentioned otherwise, bars represent the mean and standard deviation (SD). Beta-diversity (Jaccard and Yue and Clayton indices) was compared between groups using the analysis of molecular variance (AMOVA) test. Only comparisons within the same age were used because age has been demonstrated to be a major factor influencing microbiota composition.

Differences between groups were further explored using linear discriminant analysis effective size (LEfSe) [46], which uses the factorial Kruskal–Wallis sum rank and a subsequent pairwise test (Wilcoxon rank-sum test) to detect features with biological significance by comparing the abundance in all four treatment groups at different sampling days, including those with low abundance. As a last step, LEfSe uses linear discriminant analysis (LDA) to estimate the effect size of each differentially abundant feature. Alpha values for the factorial Kruskal–Wallis’s test among classes and the pairwise Wilcoxon test between subclasses were set to 0.05. The threshold on the logarithmic LDA score for discriminative features was set to 2.0.

## 3. Results

### 3.1. Animals

A total of 244 Cobb chicks hatched. After hatching, 15 and 24 newly hatched chicks in the Autoclaved and Control groups, respectively, jumped out from the incubator tray. Therefore, these birds were withdrawn from the experiment. Hence, 70, 55, 72, and 47 chicks were included in the Organic, Autoclaved, Conventional, and Control groups, respectively. The chicks received the gavage uneventfully and drank all the water with the inoculum during the first day of life.

### 3.2. Characterization of the caecal Microbiota

A total of 10,811,895 reads from 213 samples were used for the final analysis (mean: 50,760 reads per sample; SD: 12,275). The sample providing the lowest number of reads (21,146 reads) was used as a cut-off for subsampling the other samples to decrease bias caused by non-uniformity. The average coverage was 99.64% (SD, 0.30) using this cut-off, indicating that the analysis could detect almost all genera present in the caeca of the broiler chickens.

### 3.3. Donors’ Microbiota

The bacterial community members and the relative abundances of their caecal microbiota were analyzed at the genus level to explore differences in the microbiota of conventionally, organically reared donor birds and the autoclaved caecal contents (Figure 1). The inoculum obtained from organic donors contained 84 different genera; therefore, it was richer than the inoculum from conventional donors, which had 58 different genera. Unclassified Clostridiales was the most abundant taxon (15.25%) in conventionally raised broiler chickens, much higher than in the organically farmed laying hens (1.48%). The other main genera present in the conventional inoculum included unclassified Lachnospiraceae (14.32%), unclassified Ruminococcaceae (13.85%), and unclassified Firmicutes (7.89%). In the inoculum originated from organic donors, unclassified Bacteroidales (16.98%), Bacteroides (15.56%), Megamonas (11.88%), and unclassified Acidaminococcaceae (10.17%) comprised the most abundant bacteria.

### 3.4. Alpha-Diversity

Alpha-diversity was analyzed using the Chao estimator of richness, Simpson, and Shannon indices to understand the treatment’s effect on the diversity of caecal microbiota among the recipient chicks. The Chao index indicated a steady increase in species richness as chicks aged (Figure 2A). Except for day 2, colonization with the Organic inoculum was associated with higher richness than the other treatment groups throughout the trial (*p* ≤ 0.05). Diversity measured by the Simpson and Shannon indices was higher in the Conventional group at early ages (days 2 and 7; *p* ≤ 0.05) compared to the Organic group (Figure 2B,C). The Simpson index in the Control group was higher compared to the Organic one on day 42 (*p* ≤ 0.05) (Figure 2B).

### 3.5. Beta-Diversity

Beta-diversity analyses considering changes in the taxonomic composition of bacterial communities over time are depicted in Figure 3A,B. Treatment with the microbial inocula (Organic and Conventional) was associated with statistical differences in the membership and structure of the recipient chicks from day 7 to day 42. Statistical comparisons were analyzed within each age because this is a well-established factor of microbiota variance. All *p* values were ≤0.001, except for the following comparisons in Membership: Conventional–Organic Day 2 (*p* = 0.347) and Conventional–Organic Day 7 (*p* = 0.006); and in Structure: Autoclaved–Control Day 2 (*p* = 0.031), Autoclave–Conventional Day 2 (*p* = 0.002), Autoclave–Organic Day 2 (*p* = 0.002), Conventional–Organic Day 2 (*p* = 0.500), Control–Organic Day 7 (*p* = 0.01), Conventional–Organic Day 7 (*p* = 0.005), Control–Organic Day 14 (*p* = 0.037), Conventional–Organic Day 14 (*p* = 0.013), Control–Organic Day 28 (*p* = 0.009), Conventional–Organic Day 28 (*p* = 0.009), Control–Conventional, Day 42 (*p* = 0.224), Control–Organic Day 42 (*p* = 0.200) and Conventional–Organic Day 42 (*p* = 0.070).

It can be observed In the pCoA plots that the microbiota of chicks receiving the inocula (Conventional and Organic) resembled the composition of the donors from day seven onwards. Furthermore, the microbiota composition of the chickens became more similar as they aged. Even so, those who had received the Organic inoculum remained markedly different until the end of the study (day 42).

### 3.6. Recipient Chick’s Microbiota

The relative abundances of the most common genera found in each treatment group at different sampling times are presented in Figure 4. The 26 most common genera (>1% abundance) found in the samples represented, on average, 92.89%, 93.71%, 93.72%, and 94.33% of all sequences in the Control, Autoclaved, Conventional, and Organic group, respectively. The five most common genera were Bacteroides, unclassified Enterobacteriaceae, unclassified Lachnospiraceae, Faecalibacterium, and unclassified Ruminococcaceae (representing, overall, 14.45%, 11.48%, 7.93%, 6.07%, and 5.84% of all sequences, respectively).

The LEfSe analysis revealed that at day 7, six, three, and three taxa were significantly overrepresented in the microbiota of Conventional, Control, and Autoclaved recipient chicks, respectively. No specific taxa were associated with the Organic birds on day 7. There were overrepresentations of one taxon (in Organic), one taxon (in Conventional), three taxa (in Control), and four taxa (in Autoclaved) in chicks at day 14. Lastly, on day 42, one, four, three, and four taxa were significantly overrepresented in the microbiota of Organic, Conventional, Control, and Autoclaved birds, respectively. There were no significant differentially abundant features in the microbiota of the treatment groups on days 2 and 28 (Figure 5).

### 3.7. Cytokines Analyses

On day 2, birds from the Conventional group had significantly greater serum levels of IL-1β and IL-6 than Control, Organic, and Autoclaved birds (*p* ≤ 0.05). In addition, the Conventional group had a significantly higher concentration of IL-6 on day 28 than the other treatment groups (*p* ≤ 0.001). There was no treatment effect on the interleukins measured among recipient chicks on day 7, nor notable differences in IL-8 production during the trial (Figure 6).

RT-qPCR was performed to verify the results obtained from the ELISA analyses and further examine the cytokines expressed by the caecal tonsils isolated from the receipt chicks (Figure 7). The Conventional inoculum induced a significant increase in IL-4 on day two compared to birds in the Organic treatment (*p* ≤ 0.01). On day 2, the observed level of IL-6 transcript was consistent with the amount of IL-6 detected in serum (*p* ≤ 0.001). The Conventional inoculum induced significant upregulation of IL-10 on days 2, 7, and 42 (*p* ≤ 0.05, *p* ≤ 0.05, and *p* ≤ 0.01, respectively) in the treated chicks. IL-12 mRNA expression was exacerbated in the Conventional group compared to the Organic and Autoclaved groups (*p* ≤ 0.001) on day 7.

### 3.8. Mortality and Eimeria spp. Infestation

One bird per group died during the first week of life. During weeks 2 and 3, each group recorded another death, except in the Conventional group, where nine birds died (Appendix A). Post-mortem examination of dead chicks in the Conventional group revealed *Eimeria* infestation and hemorrhagic typhlitis. Treatment with amprolium (10 mL/gal) was administered only to this group from day 19 to day 25 in drinking water. One bird in the Conventional and Control groups died after the third week of the experiment.

A quantitative modified Wisconsin technique for diagnosing and differentiating intestinal *Eimeria* spp. Oocysts was carried out at the Centre de diagnostic vétérinaire de l’Université de Montréal (CDVUM) in the inocula made of the donors’ caecal contents. The analysis revealed oocysts of five different *Eimeria* spp. In the inocula given to the recipient chicks. A total of five *E. mitis* oocysts were found on the entire assessed slide in the Organic inoculum. Between 11 and 100 oocysts per microscope field (×10) were found in the Conventional inoculum: 3% from *E. acervuline*, 71% from *E. tenella*/*E. praecox*/*E. necatrix*, 17% from *E. brunetti*, and 9% from *E. maxima*. No oocysts were found in the Autoclaved inoculum.

### 3.9. Body Weight

The average body weight observed in each group is presented in Appendix A. On day 7, the Control group chicks had a significantly higher body weight than the Conventional and Organic groups. On days 14, 28, and 42, the Conventional group birds had a lower weight than the rest of the groups: Control–Conventional Day 7 (*p* = 0.0022), Control–Organic Day 7 (*p* = 0.0051), Control–Conventional Day 14 (*p* = 0.0127), Conventional–Autoclaved Day 14 (*p* = 0.0017), Conventional–Organic Day 14 (*p* = 0.0021), Control–Conventional Day 28 (*p* = 0.0098), Conventional–Autoclaved Day 28 (*p* = 0.0004), Conventional–Organic Day 28 (*p* = 0.0116), Control–Conventional Day 42 (*p* = 0.0085), Conventional–Autoclaved Day 42 (*p* = 0.00880), and Conventional–Organic Day 42 (*p* = 0.0009).

## 4. Discussion

Microbiota manipulation has recently gained a lot of attention, both in human medicine and animal production, focusing on its potential to prevent diseases and improve productivity. The objective of the present study was to test an early-life protocol of microbiota transplantation in chicks. The present study demonstrated that a caecal microbiota transplant administered on day one of life could permanently colonize the intestinal tract of chickens. Our results showed that the caecal bacterial composition resembled the donor’s microbiota from day seven after hatching and lasted until slaughter age (42 days). Noteworthy, chickens receiving the inoculum obtained from organic hens had a markedly distinguished microbiota, while other groups converged on a more similar bacterial profile.

Elokil et al. reported the stability of a transplanted faecal microbiota through four weeks of life, although there was daily inoculation throughout the experiment [47]. Metzler-Zebeli et al. also reported the persistence of a transferred microbiota during 30 days of life following inoculation on days 1, 6, and 9 [48]. Fu et al. showed phenotypic changes still present up to 16 weeks of age in laying hens after FMT administration during the first five weeks of life [49]. Yu et al. showed persistency of a transplanted microbiota from an adult donor through three weeks of life when administrated to newly hatched laying hens on days 1, 3, and 5 [50]. Similarly, the microbiota transferred in the present study persisted for six weeks after being administered only once on day one of life. Taken together, the current data suggest that the first days of life are optimal for successful microbiota manipulation in chicks. However, it is possible that such premature maturation may be accompanied by other consequences not evaluated in those studies.

Interestingly, the microbiota found in the Control and Autoclaved groups converged on a very similar microbiota to the Conventional group and, consequently, to the Conventional donor inoculum at day 42 post-hatch. This might indicate that genetic and environmental factors, such as diet and the bacteria found in wood shavings, likely influence the microbiota typically seen in commercial flocks.

In a more natural setting, chicks are expected to acquire much of their microbiota through vertical transfer from the hen. Kubasova et al. identified 13 genera whose abundance in chick’s caecal microbiota increased between 8- and 120-fold and five genera whose abundance decreased between 5- and 45-fold in chicks reared in the presence of an adult hen [51]. Furthermore, the lack of microorganism vertical transmission from the hen, cloacal, and nest onto eggs in industrial systems might predispose to colonization by environmental bacteria [52]. As such, the conventionally raised donor’s microbiota was likely acquired from the environment, explaining why the Control or Conventional group converged towards a similar microbiota composition and structure. It has also been demonstrated that resistance to caecum colonization by Salmonella enteritidis was increased by more than 5-log for chicks reared in contact with an adult hen, indicating that adequate vertical transmission of the microbiota might have a protective effect against pathogens [51].

A group receiving an inoculum similar to that administrated to the Organic group but which did not contain any viable organism (Autoclaved group) was included in the study in order to investigate whether it would be possible to modulate the intestinal microbiota of chickens based on the nutrients and metabolites present in that ecosystem, as it has been suggested [53]. This could overcome the undesired risks associated with faecal or caecal microbiota transplantation, such as pathogen transmission, donor selection and screening, and inoculum stability and viability after collecting caecal contents. In the present study, the administration of autoclaved caecal content failed to modulate the caecal microbiota of the recipients, as it resembled the microbiota of Controls during the study period. Unexpectedly, the autoclaving process appears to be responsible for DNA degradation, as indicated in Figure 1, which might have been accompanied by the degradation of essential compounds in the solution. Unfortunately, the autoclaved solution used in the gavage was not sequenced, only the diluted solution in the drinking water given on the first day of life.

It has been suggested that the microbial composition of the intestinal microbiota influences poultry performance [54], likely due to some bacterial species having the capacity to improve feed conversion, at least in part, by increasing the production of volatile fatty acids (VFA) [55,56]. Microbial transplantation in chickens has been attempted to promote growth performance, with inconsistent results [47,49,50]. The present study expected that colonization with a highly diverse microbiota containing lactic acid bacteria (i.e., *Lactobacillu*s spp.) and VFA producers (i.e., *Bacteroides* spp.) could be associated with significant average weight gain in the Organic group. However, animals in the Conventional and Organic groups had significantly less weight gain than the Control group on day 7. Although the Organic group had a significantly higher weight than the Conventional group on day 14, the importance of this finding is uncertain because the number of animals per group used in the present study precludes a robust analysis of zootechnical parameters.

The Conventional group had significantly higher richness (Chao index) on day 2, but the Organic group had greater richness for the rest of the study. Conversely, the Organic group presented significantly lower diversity at day 42 (Shannon and Simpson indices), demonstrating that although those chickens harboured many different species in their caecum, those populations were unevenly distributed, dominated by a few taxa, as shown in Figure 4.

One of the main disadvantages of microbiota transplantation is the risk of transmitting harmful microorganisms such as enteropathogenic bacteria, viruses, and parasites [57]. The unusual mortality in the Conventional group during the second week of the experiment was attributed to infestation with *Eimeria* spp., whose origin was later confirmed from the Conventional inoculum. This event emphasizes the need to screen for *Eimeria* and other pathogens in donor caecal contents, which implies time-consuming delays before transplantation [58]. Interestingly, *Eimeria* spp. was found at a lower concentration in the Organic donor’s inoculum but did not cause clinical disease in recipient chickens. Whether this was caused by an insufficient infectious dose or by a protective effect caused by the Organic microbiota is beyond the scope of this study to investigate. In addition, it is possible that the coccidiosis observed in the Conventional group could have affected weight gain and inflammatory cytokines levels in that group [36,59,60,61]. Even so, neither the infestation nor treatment caused marked changes in the intestinal microbiota composition.

Different microbiota profiles can be associated with varying degrees of inflammation in various species, including chickens [54,62,63,64]. Interestingly, it has been shown that microbiota transplantation can positively impact intestinal morphology, increasing the intestinal villi height and villus height/crypt depth ratio [49,50]. The early-life increase in IL-4 mRNA expression in the Conventional group is challenging to interpret, as IL-4 can mediate macrophage and lymphocyte activation, leading to a state of inflammation that could affect intestinal immune function and maturation [65]. Nevertheless, it is uncertain if this result was caused by an inflammatory-like microbiota profile present in the inoculum obtained from Conventional chickens or by the presence of *Eimeria* spp. In-ovo IL-4 plasmid injections concomitant with coccidia vaccination have been shown to modulate macrophage nitric oxide production and reduce CD8^+^ T-cell and lymphocyte proliferation in caecal tonsils while reducing the impact on body weight in birds challenged with *Eimeria* spp. [66]. Therefore, an early IL-4 response may be linked to a pre-hatching exposure to the parasite. Along the same lines, the present study also reports increased inflammatory markers in the Conventional group chickens: production of IL-1β and IL-6 in plasma on days 2 and 28, increased IL-6 expression in the caecal tonsils at days 7 and 42, and increased IL-12 expression at seven days of life. These findings are all consistent with an *Eimeria* spp. infestation, and thus the condition prevents us from making more profound interpretations of these findings [67].

Conversely, there was a higher expression of the anti-inflammatory IL-10 in the Conventional group’s caecal tonsils at days 7 and 42. While it follows what was previously seen by Metzler-Zebeli et al. [48], who reported increased IL-10 expression following microbiota transplantation from what was considered a “good” donor (in this case, a highly feed-efficient broiler), the IL-10 increase seen here may be due to the *Eimeria*. IL-10 is primarily an anti-inflammatory cytokine, and several pathogens rely on its production for pathogenesis [61,68]. As such, since the Conventional group received both a transplant from commercial broilers and an *Eimeria* infestation, we cannot say if the increased IL-10 expression is due to the infestation or the transplant. A delicate balance between pro-inflammatory and anti-inflammatory processes is required in order to allow for an adequate immune response to pathogens and protect the animal from excessive inflammation. Microbiota transplantation has been used to promote intestinal immunity, maturation, and growth performance [48,50]. However, further studies involving pathogen challenges are required in order to investigate the protective potential of microbiota profiles originating from wild birds or organically raised chickens.

Although the microbiota transplantation protocol adopted in this study successfully colonized the intestinal tract of chickens up to slaughter age, it is not feasible on an industrial scale since individual inoculation is labour-intensive. Still, bacteriotherapy administered in drinking water could be an alternative. Regardless, obtaining sufficient caecal content from certified pathogen-free flocks is also unrealistic. From this research and the work of others [49], it appears that donor selection is the most crucial step in determining the benefits of microbiota transplantation, and the importance of pathogen screening cannot be understated. Nevertheless, the study provides new information on microbiota manipulation methods and the interaction between the intestinal microbiota and chickens.

## 5. Conclusions

Microbiota transplantation using different microbiota profiles persistently colonized the intestinal tract of newly hatched broiler chicks. Colonization with an Organic microbiota was associated with higher richness but lower diversity and marked compositional differences compared to Control and Conventional microbiota profiles. Chicks colonized with the microbiota of conventionally raised broilers had more elevated pro- and anti-inflammatory cytokines, but coccidiosis could be a confounding factor. Future studies are necessary in order to evaluate the importance of microbiota composition during infections with common enteropathogens affecting the poultry industry (i.e., *Eimeria* spp. and *Clostridium perfringens*). This study highlights the importance of a strict screening protocol for the presence of pathogens in the donors’ intestinal content in preventing the passing along of pathogens or other unwanted microorganisms.

## Figures and Tables

**Figure 1 animals-13-02633-f001:**
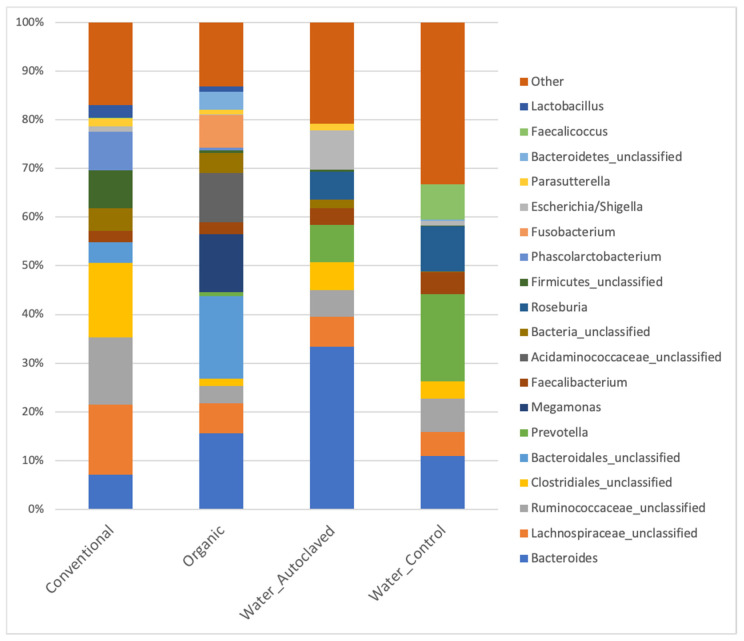
Relative abundance of the main bacteria genera present in the donor chickens (Conventional and Organic) and in the water administered to Control and Autoclaved groups.

**Figure 2 animals-13-02633-f002:**
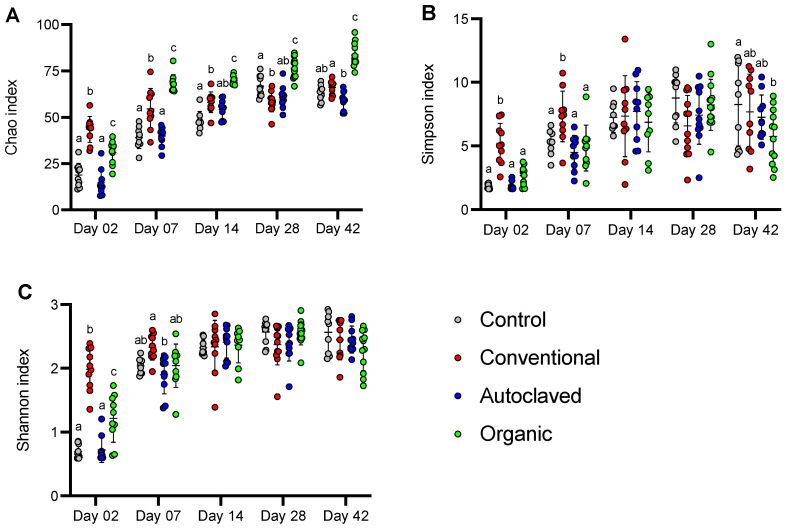
Alpha-diversity indices of caecal microbiota of chicks. (**A**) Chao, (**B**) Simpson, and (**C**) Shannon indices of caecal microbiota communities of chicks inoculated with water (Control), caecal content from conventionally raised broilers (Conventional), caecal content from organically raised laying hens (Organic), or autoclaved caecal content from organically raised laying hens (Autoclaved) at day 2, 7, 14, 28, and 42 post-hatch. For the same day, groups designated with a different letter are statistically different from one another (*p* ≤ 0.05). Statistical analysis was performed using the mixed-effect model and Tukey’s multiple comparison tests. Bars represent mean and SD.

**Figure 3 animals-13-02633-f003:**
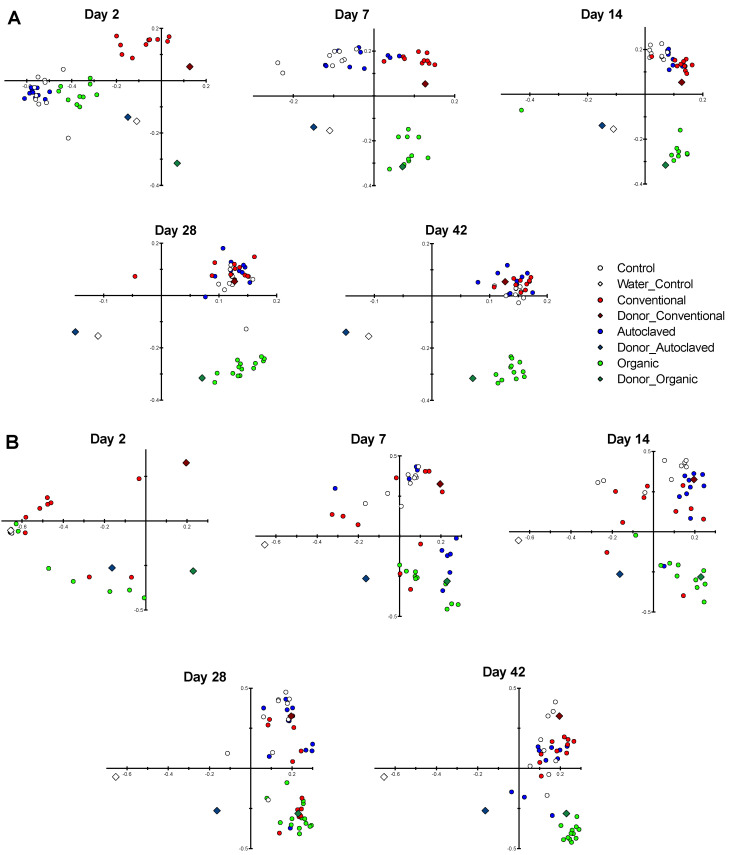
Bidimensional principal coordinate analysis (pCoA) representation of bacterial communities in caecal contents of the recipient chicks on days 2, 7, 14, 28, and 42. (**A**) The community membership, which considers the presence or absence of each taxon addressed by the Classic Jaccard index. (**B**) The structure, which also considers the abundance of each taxon, as addressed by the Yue and Clayton analysis. Dots represent the microbiota of recipient chickens, and diamonds represent donors. Organic (green): free-range laying hens without antibiotics; Autoclaved (blue): autoclaved caecal contents of organically reared laying hens; Conventional (red): factory-raised broiler chickens collected from the slaughterhouse; Controls (white): received only water.

**Figure 4 animals-13-02633-f004:**
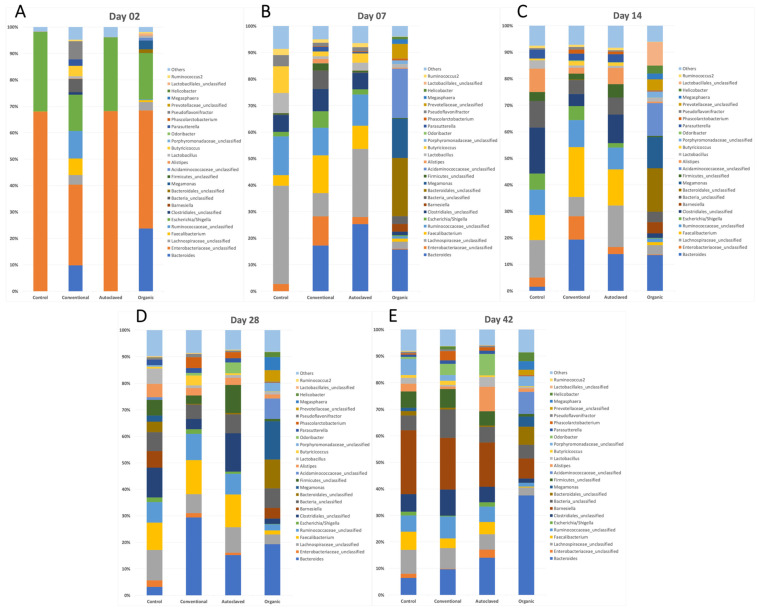
Composition of the bacterial communities in the recipient chicks. Relative abundance of the most abundant bacterial genera (>1% abundance) in the caecal contents of the newly hatched chicks at days 2 (**A**), 7 (**B**), 14 (**C**), 28 (**D**), and 43 (**E**).

**Figure 5 animals-13-02633-f005:**
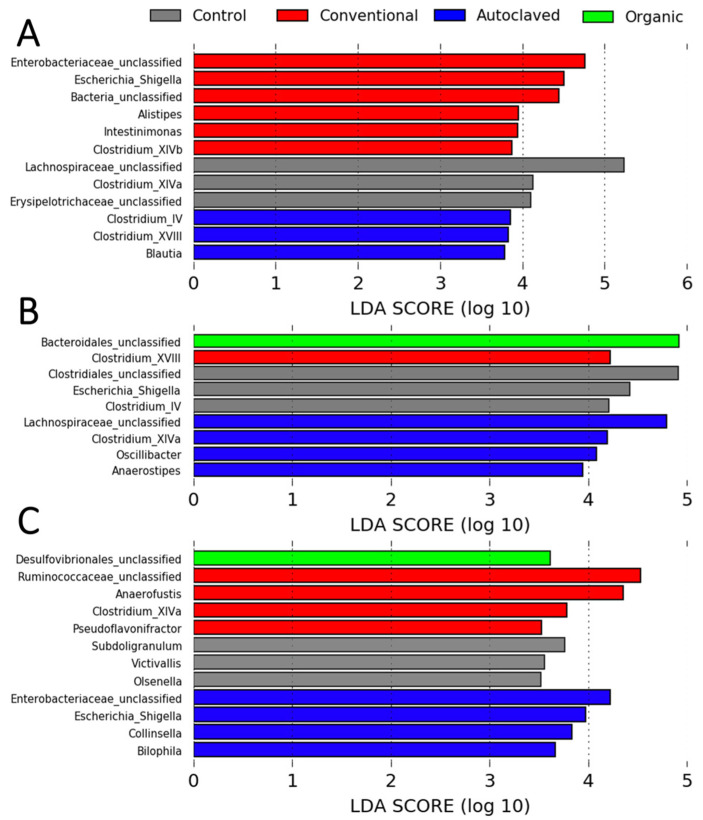
Linear discriminant analysis effect size (LEfSe) showing taxa that were significantly differentially abundant in the microbiota of the treatment groups on days 7 (**A**), 14 (**B**), and 42 (**C**). All LDA scores > 2. LDA, linear discriminant analysis.

**Figure 6 animals-13-02633-f006:**
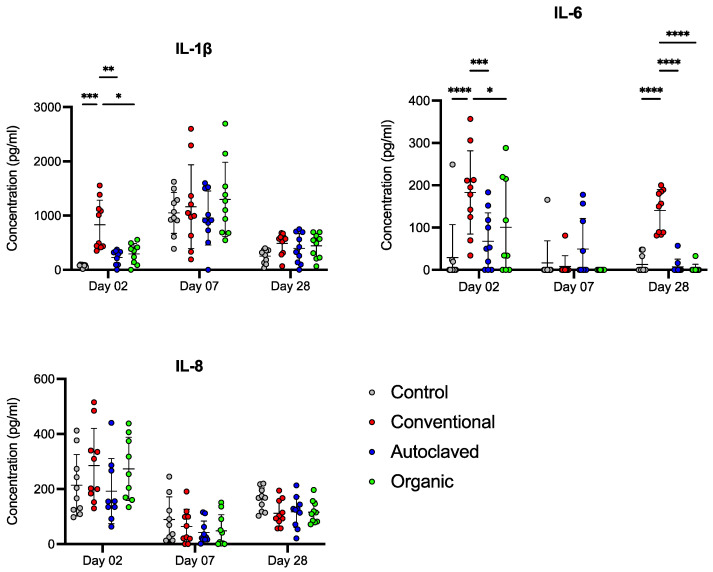
IL-1β, IL-6, and IL-8 concentrations in serum determined by chicken-specific-antigen ELISA on days 2, 7, and 28. Statistical analysis was performed using Two-way ANOVA followed by Tukey’s multiple comparison tests. Bars represent mean and SD. Results were considered statistically significant if *p* ≤ 0.05. * *p* ≤ 0.05; ** *p* ≤ 0.01; *** *p* ≤ 0.005; **** *p* ≤ 0.001.

**Figure 7 animals-13-02633-f007:**
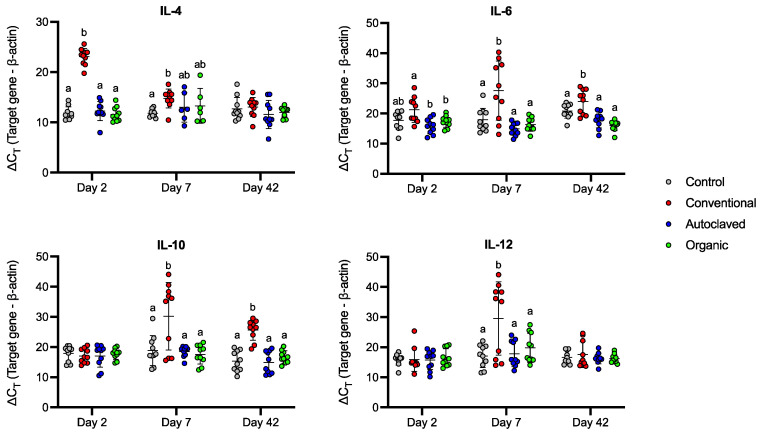
Relative mRNA expression of cytokines IL-4, IL-6, IL-10, and IL-12 in the recipient chicks’ caecal tonsils on days 2, 7, and 42. Data are shown as the difference between the target gene and β-actin expression. Gene expression among treatment groups was compared statistically using one-way ANOVA followed by Tukey’s multiple comparison tests. Bars represent mean and SD. For the same day, groups designated with different letters are statistically different from one another (*p* ≤ 0.05).

## Data Availability

Data will be provided upon request.

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
