# Peer review of "Colonization of the Gastrointestinal Tract of Chicks with Different Bacterial Microbiota Profiles"

_animals, 2023, doi:10.3390/ani13162633_

Round 1

Reviewer 1 Report

Dear authors:

I consider that your paper will be very useful to increase the knowledge of the early gut colonization with defined inoculum in chickens and the role of different microbiota components in the mucosal maturation, including gut associated immune system.

During the trial you have a serious trouble related with the contamination of the inoculum used for conventional experimental group. This contamination could produce some difficulties to infer some significant points of your results, specially for some immunological parameters that were significantly different in the conventional experimental group. Nevertheless, in my opinion, the infestation with Eimeria didn`t have adverse effects on caecal microbiota transplantation as can be deduced for the high microbiota profile similarity between the Conventional, the Autoclaved, and the Control groups. This profile similarity could be associated with the normal chicken colonization with the conventional microbiota related with the conventional conditions of poultry production in the geographical area where the experiment was conducted.

It is very important to remark your conclusion about the need for a strict screening protocol for the presence of pathogens in the donors, I believe that this screening is essential to use the microbiota transplantation at field conditions.

Additionally, I believe that you must better explain the apparent discrepancy between the number of eggs included in the initial constitution of experimental groups (75 eggs per experimental group) and the number of hatched chickens at each experiment group: 70, 55, 72 and 47 chicks in the Organic, Autoclaved, Conventional, and Control group. You should include some additional information that help to explain the significant reduction in the number of chicks in the Autoclaved and Control groups.

Additional minor comments:

- In Figure 2B, the letter of the statistical result corresponding to the Organic group on day 7 is not included.

- Please, use the term “infestation” when you refer to the presence of Eimeria in the gut of animals and on the inoculum.

Author Response

We greatly appreciate your time and suggestions to significantly improve our manuscript. We are glad to inform that all your comments have been incorporated into the new version of the manuscript, that we hope is now suitable for publication. Please see below our answers to each of your concerns.

Best regards.

Reviewer 1:

Dear authors:

I consider that your paper will be very useful to increase the knowledge of the early gut colonization with defined inoculum in chickens and the role of different microbiota components in the mucosal maturation, including gut associated immune system.

During the trial you have a serious trouble related with the contamination of the inoculum used for conventional experimental group. This contamination could produce some difficulties to infer some significant points of your results, especially for some immunological parameters that were significantly different in the conventional experimental group. Nevertheless, in my opinion, the infestation with Eimeria didn`t have adverse effects on caecal microbiota transplantation as can be deduced for the high microbiota profile similarity between the Conventional, the Autoclaved, and the Control groups. This profile similarity could be associated with the normal chicken colonization with the conventional microbiota related with the conventional conditions of poultry production in the geographical area where the experiment was conducted.

It is very important to remark your conclusion about the need for a strict screening protocol for the presence of pathogens in the donors, I believe that this screening is essential to use the microbiota transplantation at field conditions.

Answer: Thank you very much for your comments. The last sentence of the conclusion highlights the importance of donor screening: “This study also highlights the need for a strict screening protocol for the presence of pathogens in the donors’ intestinal content”. This is also emphasized in the discussion.

Additionally, I believe that you must better explain the apparent discrepancy between the number of eggs included in the initial constitution of experimental groups (75 eggs per experimental group) and the number of hatched chickens at each experiment group: 70, 55, 72 and 47 chicks in the Organic, Autoclaved, Conventional, and Control group. You should include some additional information that help to explain the significant reduction in the number of chicks in the Autoclaved and Control groups.

Answer: The discrepancy between the number of birds per group was caused by 15 and 24 chicks in the Autoclaved and Control groups, respectively, that jumped out from the egg incubator tray after hatching. Hence, these birds were withdrawn from the experiment to avoid possible contamination. This information has been added to the first part of Results.

Additional minor comments:

- In Figure 2B, the letter of the statistical result corresponding to the Organic group on day 7 is not included.

Answer: Thank you for noticing that. The letter has been included.

- Please, use the term “infestation” when you refer to the presence of Eimeria in the gut of animals and on the inoculum.

Answer: The term “infection” has been replaced.

Reviewer 2 Report

This manuscript provides valuable information on the ability of fecal microbiota transplantation to produce long-term modifications in the microbiome of naive chicks. However, there are a few issues with the manuscript that must be clarified or changed prior to acceptance. 

Introduction: Authors argue that the microbiome of conventional chicks at hatch is "abnormal" and "unnatural" due to the lack of interaction with hens and the sterilized environment potentially leaving birds open to colonization with pathogens. However, commercial embryos are known to possess their own dynamic microbiome prior to pipping and interaction with the egg shell or environment which is most likely due to maternal transfer during egg formation. To my knowledge, no work has been done to investigate the day-of-hatch or embryo microbiome in eggs that had maternal contact throughout incubation. Therefore, the argument that the microbiome of hatchlings in a commercial setting are unnatural, abnormal, or bad is unfounded. We can speculate that it develops differently during the first week of life but not that the initial microbiome at hatch is different. 

Section 2.3: Did the organic and autoclaved inocula come from two separate sets of 10 birds? Or were cecal contents collected from 10 birds, homogenized, and split into two 20 g aliquots? Currently it is unclear. 

L212-213: Was AMOVA run on all of the data together or was the data split by age prior to analysis? More clarification is needed on the model used here. 

Results: It is concerning that the autoclaved donor material more closely clusters with the water than the organic donor in the beta diversity analysis. While the autoclaved group is not expected to have live bacteria, it should still possess microbial DNA that would show up in sequencing. Previous work has shown that 121C for 80 minutes is necessary to remove amplifiable DNA template. As the material for autoclaving was taken from the same, or very similar, birds as the organic group, it would make more sense for the autoclaved material to more closely align with it than the sterile water. Please explain your rationale as to the validity of your results and consider adding the composition of the autoclaved material to figure 1.

Beta Diversity: You mentioned in the statistics portion that you ran ANCOM to test for statistical diffferences but no statistics are shown in figure 3 or in the results section. Please add to both. 

Author Response

Reviewer 2:

This manuscript provides valuable information on the ability of fecal microbiota transplantation to produce long-term modifications in the microbiome of naive chicks. However, there are a few issues with the manuscript that must be clarified or changed prior to acceptance.

Introduction: Authors argue that the microbiome of conventional chicks at hatch is "abnormal" and "unnatural" due to the lack of interaction with hens and the sterilized environment potentially leaving birds open to colonization with pathogens. However, commercial embryos are known to possess their own dynamic microbiome prior to pipping and interaction with the eggshell or environment which is most likely due to maternal transfer during egg formation. To my knowledge, no work has been done to investigate the day-of-hatch or embryo microbiome in eggs that had maternal contact throughout incubation. Therefore, the argument that the microbiome of hatchlings in a commercial setting are unnatural, abnormal, or bad is unfounded. We can speculate that it develops differently during the first week of life but not that the initial microbiome at hatch is different.

Answer: We agree with the reviewer. In fact the study has been developed based on the fact that the microbiome of the eggshell is different in nature compared to industrial practices as it has been demonstrated. The study aimed to investigate the impact of colonization with different microbiome profiles, but more research is required to investigate the impact of the shell microbiome in the colonization of chicks. This introduction has been rephrased as requested.

Section 2.3: Did the organic and autoclaved inocula come from two separate sets of 10 birds? Or were cecal contents collected from 10 birds, homogenized, and split into two 20 g aliquots? Currently it is unclear.

Answer: Caecal contents from all 20 organic hens were pooled together and split into two 20 g aliquots. The wording has been changed to improve clarity.

L212-213: Was AMOVA run on all of the data together or was the data split by age prior to analysis? More clarification is needed on the model used here.

Answer: The AMOVA test does only pairwise comparisons (which is a major limitation of the test). The test was applied to the whole dataset, but because age is a major factor influencing microbiota composition, only the comparisons within the same age were selected. That information has been added to the text.

Results: It is concerning that the autoclaved donor material more closely clusters with the water than the organic donor in the beta diversity analysis. While the autoclaved group is not expected to have live bacteria, it should still possess microbial DNA that would show up in sequencing. Previous work has shown that 121C for 80 minutes is necessary to remove amplifiable DNA template. As the material for autoclaving was taken from the same, or very similar, birds as the organic group, it would make more sense for the autoclaved material to more closely align with it than the sterile water. Please explain your rationale as to the validity of your results and consider adding the composition of the autoclaved material to figure 1.

Answer: As the reviewer mentioned, we did not expect to see complete degradation of DNA after autoclaving. As explained in the discussion, that was in fact the rationale of adding this experimental group in the study, to have parts of intact proteins that could be able to induce an immune response without the threat of transmitting pathogens or pathobionts. The composition of the solution given to Autoclaved group was not sequenced, only the dilution given to day-old chicks. The bacterial composition of this solution as well as from the water given to controls were added to Figure 1. The unexpectedness of this result has also been added to the discussion:

“Unexpectedly, it seems that autoclaving the inoculum was responsible for DNA degradation as indicated in Figure 1, that might have been accompanied by degradation of essential compounds in the solution. Unfortunately, the autoclaved solution used in the gavage was not sequenced, only the solution diluted in water giving in the drinker during the first day.”

Beta Diversity: You mentioned in the statistics portion that you ran ANCOM to test for statistical diffferences but no statistics are shown in figure 3 or in the results section. Please add to both.

Answer: We believe the reviewer refers to the AMOVA test. In fact results of the comparisons were missing and were included in the results (section 3.5).

Reviewer 3 Report

The manuscript describes the experimental procedures and results of a trial, which appears to be well designed and performed. Results are will be of a high interest for readers of the journal.

There are only minor comments or suggestions from my side

Line 59.- highest feed efficiency (lowest feed conversion ratio)

Line 99.- the experimental design also includes the supply of conventional microbiota vs a negative control (closer to the commercial hygienic measures), which would suggest an additional hypothesis.

Line 358.- Considering that only 2 replicates are used per treatment, showing statistical analyses of BW is questionable (see Line 429). I would suggest better to delete figure 8 and provide average BW values in the text.

In general, please keep the same order among treatments for the different figures (for example, Figure 2: Control/Conventional/autoclaved/organic; and Figure 4 : Control/Autoclaved/Conventional/Organic, and so on

Author Response

Reviewer 3:

The manuscript describes the experimental procedures and results of a trial, which appears to be well designed and performed. Results are will be of a high interest for readers of the journal.

There are only minor comments or suggestions from my side

Line 59.- highest feed efficiency (lowest feed conversion ratio)

Answer: Thank you for your correction.

Line 99.- the experimental design also includes the supply of conventional microbiota vs a negative control (closer to the commercial hygienic measures), which would suggest an additional hypothesis.

Answer: We have rephrased the hypothesis to cover this concern.

Line 358.- Considering that only 2 replicates are used per treatment, showing statistical analyses of BW is questionable (see Line 429). I would suggest better to delete figure 8 and provide average BW values in the text.

Answer: We agree with the reviewer’s comment and have moved the figure as supplementary material.

In general, please keep the same order among treatments for the different figures (for example, Figure 2: Control/Conventional/autoclaved/organic; and Figure 4 : Control/Autoclaved/Conventional/Organic, and so on

Answer: All figures have been organized as per the reviewer’s suggestion.